# Alopecia Areata and Dexamethasone Mini-Pulse Therapy, A Prospective Cohort: Real World Evidence and Factors Related to Successful Response

**DOI:** 10.3390/jcm11061694

**Published:** 2022-03-18

**Authors:** Manuel Sánchez-Díaz, Trinidad Montero-Vilchez, Ahinoa Bueno-Rodriguez, Alejandro Molina-Leyva, Salvador Arias-Santiago

**Affiliations:** 1Dermatology Unit, Hospital Universitario Virgen de las Nieves, IBS Granada, 18002 Granada, Spain; manolo.94.sanchez@gmail.com (M.S.-D.); tmonterov@gmail.com (T.M.-V.); abuenorodriguez@hotmail.com (A.B.-R.); salvadorarias@ugr.es (S.A.-S.); 2Trichology Clinic, Hospital Universitario Virgen de las Nieves, 18002 Granada, Spain; 3Dermatology Department, School of Medicine, University of Granada, 18002 Granada, Spain

**Keywords:** alopecia areata, mini-pulse therapy, corticosteroids, dexamethasone

## Abstract

The mini-pulse oral corticosteroids treatment for alopecia areata (AA) is an accessible treatment whose efficacy and adverse effects have not yet been properly described. The objective of the study was to assess the effectiveness and safety of the mini-pulse oral corticosteroids treatment in patients with AA, and to explore potential factors associated to the response in a real-world setting. An observational study of a prospective cohort to assess the effectiveness and safety of a mini-pulse dexamethasone treatment in patients with AA, who failed to improve with topical therapies, was performed. A SALT II score and other clinical and safety variables were recorded at baseline, 3, 6, 9, and 12 months. An overall significant and progressive decrease of the SALT score was found during treatment: SALT-50 response was achieved after 9 months in 51.8% of the patients. Hypothyroidism and early age of onset were predictors of the lack of response to treatment. The treatment combination with oral minoxidil showed no effect on the SALT score reduction. Dexamethasone daily and cumulative dose were associated with a higher percentage of side effects. To conclude, the mini-pulse oral corticosteroids treatment is an effective treatment for AA, although patients with an early onset of the disease and hypothyroidism may not benefit.

## 1. Introduction

Alopecia areata (AA) is a common inflammatory and non-scarring type of alopecia, which affects patients of all ages, sexes, and ethnicities. The severity of the disease varies from mild cases, with small well-circumscribed patches of hair loss; to severe cases of generalized alopecia of scalp and body [1]. As hair loss might have an impact on self-image, it has been proven that AA is related to great psychological impairment [2,3,4].

On the other hand, the etiology of AA is not completely understood yet, although it is thought that the loss of immune privilege in the hair follicle could lead to an autoimmune hair follicle damage that is responsible for alopecia [5]. More specifically, it is thought that CD8+ T cells are involved in the release of interferon-γ, which leads to the disruption of the immune tolerance of hair follicles and the subsequent exposure of self-antigens [6].

A wide variety of treatments have been used in the management of AA [5,7]. Topical or intralesional therapies, including corticosteroids, anthralin, dyphencyprone, and minoxidil, are mainly used in mild, less extensive cases [8]. Extensive cases of AA, such as AA totalis or universalis, are more commonly treated with systemic drugs [9], including oral corticosteroids [10] or immunosuppressive agents, such as methotrexate or azathioprine [11,12]. More recently, Janus kinase inhibitors (JAKi) have emerged as alternatives for the treatment of severe cases [13], which act through the intracellular interruption of the JAK-STAT pathway. Finally, the use of advanced therapies, such as mesenchymal stromal cells, could be a promising field for the future treatment of alopecia areata [14].

Given the accessibility and low price, the use of oral corticosteroids continues to be an option of choice in moderate to severe cases of alopecia areata. Oral corticosteroids, administered in pulses, have shown to be effective in the management of AA [15,16], although most studies address the issue of high-dose systemic corticosteroids [17,18,19,20], which can lead to several adverse effects, such as weight gain, osteopenia/osteoporosis, worse glycemic control, hyperglycemia, high blood pressure, or cataracts [21]. Mini-pulse therapy could decrease the rate of adverse effects while maintaining the efficacy of the treatment. However, real-world evidence about this treatment is still scarce, and there is a lack of studies addressing clinical and epidemiological factors associated with a satisfactory response. Therefore, the aim of the present study is to assess the effectiveness and safety of mini-pulse oral corticosteroids treatment in patients with severe AA, and to evaluate potential clinical and epidemiological factors associated to the positive response to the treatment in a real-world setting.

## 2. Materials and Methods

Design: An observational study of a prospective cohort was performed to assess the effectiveness and safety of mini-pulse dexamethasone treatment in patients suffering from alopecia areata who failed to respond to topical therapies, and to evaluate potential clinical factors associated with the response in a real-world setting. The study was approved by the Institutional Review Board of the Hospital Universitario Virgen de las Nieves.

Patients: Patients included in the study receive health care in the Trichology Clinic of the Virgen de las Nieves University Hospital. The patients are part of a prospective cohort, in which socio-demographic, clinical, and analytic data are collected. Patients were treated in the trichology consultations between January 2020–August 2021.

Inclusion criteria: The inclusion criteria were as follows: (a) Patients with clinical diagnosis of alopecia areata; (b) patients suffering from AA who were candidates to dexamethasone treatment and who failed to respond to previous topical therapy, including, at least, topical/infiltrated corticosteroids for a minimum period of 3 months; (c) SALT score > 20%; or >10% if the patient self-reported significant impairment in quality of life; (d) dexamethasone doses < 5 mg/day, two days per week.

Exclusion criteria: The exclusion criteria were: (a) The fefusal from the patient or legal representative to participate in the study; (b) patients suffering from AA who were not candidates for the dexamethasone treatment.

### 2.1. Variables of Interest

#### 2.1.1. Main Variables

The patients were assessed the day of the initiation of dexamethasone treatment and subsequently every 3 months until the treatment was discontinued or changed. The main variables included those related to the severity of the disease, and several clinical and socio-demographic factors:Severity of the disease:
○Severity Alopecia Tool II (SALT II) [22] was used to assess the severity of the disease. The variable was assessed during a routine clinical examination at baseline, 3, 6, 9, and 12 months, or at the last follow-up visit if the treatment was discontinued before 12 months. ○The presence of alopecia of eyebrows and eyelashes, and their recovery during dexamethasone treatment, were recorded.Clinical variables:
○Age of onset of the disease and evolution time of the disease at the start of the treatment were recorded. Age of onset described the age of the first episode of AA; whereas the evolution time of the disease refers to the total time between the age of onset and the age at the start of the treatment. The age of onset was classified as “early” onset (<15 years old) or “late” onset (>15 years); whereas the evolution time at the start of the treatment was classified as “early” treatment (<5 years) or “late” treatment (>5 years). The choice of cut-off points for the age of onset was made in accordance with previous reports [23,24].○Dosage schedule of dexamethasone: Patients received mini-pulse therapy. Although there is a lack of a standardized definition, most reports [25,26] addressing this term are based on the use of dexamethasone equivalent doses of less than 5 mg/day, two days a week.Safety variables: Include the appearance or worsening of hyperglycemia, high blood pressure, increasing weight, and any other relevant adverse event that could be related to the treatment of dexamethasone. These variables were assessed every three months during protocolized follow-up consultation.

#### 2.1.2. Other Variables

Socio-demographic, biometric, and clinical variables, including age, sex, and comorbidities (including hypothyroidism, diabetes mellitus, celiac disease, and other autoimmune disorders), and previous treatments for AA, were recorded in the clinical interview and physical examination.

### 2.2. Statistical Analysis

Descriptive statistics were used to evaluate the characteristics of the sample. The Shapiro–Wilk test was used to assess the normality of the variables. Continuous variables are expressed as mean and standard deviation (SD). Qualitative variables are expressed as relative and absolute frequency distributions. The χ^2^ test or Fisher’s exact test, as appropriate, were used to compare nominal variables, and the Student’s *t*-test or Wilcoxon-Mann–Whitney test were used to compare between the nominal and continuous data. To explore the possible associated factors, a simple linear regression was used for continuous variables. The β coefficient and SD were used to predict the log odds of the dependent variable. Significantly associated variables (*p* < 0.05) or those showing trends towards the statistical significance (*p* < 0.20) were included in the multivariate analysis. Multivariate logistic regression analyses were carried out to identify the factors associated with target variables. Multiple ANOVA tests (MANOVA) were used to assess the evolution of the disease severity indexes throughout the treatment. Statistical significance was considered if *p*-values were less than 0.05. Statistical analyses were performed using the JMP version 9.0.1 (SAS institute, Cary, NC, USA).

## 3. Results

### 3.1. Sociodemographic and Clinical Features of the Sample

Forty patients were included in the study. The mean age was 32.28 years (SD 16.58), with a majority of female patients (57.5% female vs. 42.5% male). The mean age of onset of the disease was 27.12 years old (SD 17.07), with 35% of patients with an early onset of the disease, and a 65% of them with a late onset. The mean evolution time of the disease at the start of the treatment was 5.11 years (SD 7.11). Previous treatments and comorbidities are described in Table 1.

The basal SALT score was 70.78% (SD 31.13) and eyebrows/eyelashes were involved in 56.4% (22/38) of the patients.

### 3.2. Characteristics of the Treatment with Dexamethasone

All patients were treated with mini-pulses of dexamethasone (Table 1). Mean dexamethasone dose was 2.72 mg/day, two days a week (SD 1.48), with a mean total treatment time of 12.22 months (SD 8.89). The majority of the patients were treated combining with proton pump inhibitors (70% -28/40-) and calcium/vitamin D supplements (82.5% -33/40-). A subset of 27.5% patients (11/40) were also treated with oral minoxidil: the doses of oral minoxidil ranged between 0.5–1 mg/day for women and 2.5–5 mg/day for men. None of the patients took any treatment other than those described above.

### 3.3. Evolution of SALT Scores during Dexamethasone Treatment

Basal SALT score and SALT scores corresponding to 3, 6, 9, and 12 months can be seen in Figure 1. A significant decrease in SALT scores was observed when analyzing the overall sample (*p* = 0.0089).

However, when a stratified analysis was performed for the age of onset, it was found that the decrease of SALT scores was not uniform between groups. Even though there were no significant differences in basal characteristics between the two groups (Table 2), “early onset” patients did not show a decrease in SALT scores over time (*p* = 0.44); whereas, “late onset” patients did (*p* < 0.01). The evolution of SALT scores in these groups can be seen in Figure 2.

Similar stratified analyses were performed for hypothyroidism and latency time for the treatment (Figure 3). No relevant results were found. The percentage of patients achieving SALT reduction can be seen in Figure 4.

Regarding eyelashes/eyebrows involvement, 63.63% (14/22) of the patients showed improvement of the alopecia at these zones.

### 3.4. SALT Reduction and Associated Factors

The SALT reduction at 3- and 9-month variables were analyzed to explore possible associated factors (Table 3). Regarding the SALT reduction at 3 months, a univariate and multivariate analysis were performed and the age of onset was found to be positively correlated with an improvement in SALT scores at 3 months (*p* = 0.02). Moreover, the female sex was related to higher decreases in SALT scores (*p* = 0.05), and hypothyroidism was found to be inversely correlated to the improvement of the SALT score (*p* = 0.03). Regarding the SALT reduction at 9 months, the age of onset was found to be related to better SALT reductions at 9 months (*p* = 0.02).

The basal SALT score was found to be unrelated to the SALT reduction at 3 months and at 9 months. Moreover, the oral minoxidil co-treatment was included in the multivariate analyses to avoid a confusion bias, and it was found to be unrelated to SALT reduction at 3 months (*p* = 0.68) and at 9 months (*p* = 0.88) (Table 3).

### 3.5. Safety of Mini-Pulses of Oral Dexamethasone and Associated Factors

A total of 40% (16/40) patients suffered from any adverse effect related to the dexamethasone treatment. None of the adverse effects were severe. These included: Weight gain, present in 35% (14/40) and glycemic disorders, present in 5% (2/40), which were the most common. Less common side effects included: acneiform rash, hirsutism, anxiety and insomnia, which were present in less than 5% (1/38) of patients. Osteopenia or osteoporosis were found in 12.5% (5/40) patients during the dexamethasone treatment.

#### 3.5.1. Overall Side Effects, Osteopenia, and Associated Factors

Regarding the overall side effects, no association was found for sex, age, age of onset, or oral minoxidil co-treatment (Table 4). However, patients with adverse effects were found to have received higher doses of dexamethasone (*p* = 0.04). The treatment time and cumulative doses of dexamethasone were close to statistical significance.

Regarding osteopenia osteoporosis, patients who suffered from osteopenia or osteoporosis during dexamethasone treatment (12.5%, 5/40) were preferably female (*p* = 0.01), had longer treatments (*p* = 0.001), and had higher cumulative doses of dexamethasone (*p* = 0.03) when compared to patients who did not suffer from it.

#### 3.5.2. Weight Gain-Associated Factors

Cumulative doses of dexamethasone were found to be related to weight gain in the univariate analysis (Table 4). However, after the multivariate analysis, no factor was significantly related to weight gain, indicating a possible confusion bias.

## 4. Discussion

Oral dexamethasone, administered as mini-pulse therapy, two days per week, is an effective and safe treatment for AA. Some specific groups, such as those with an early onset of the disease, have poorer responses, whereas those patients with a later onset have a more satisfactory improvement of SALT scores. Female sex seems to be related to early satisfactory responses to dexamethasone, and hypothyroidism seems to lead to poorer responses at the early stages of the treatment. However, these effects seem to disappear over the time. Adverse effects of treatment are common. The main factor related to the appearance of adverse effects is the dexamethasone dose, although treatment duration and cumulative doses seem to be other potential factors involved.

The corticosteroid pulse therapy has been used for many years [17,18] with variable response rates. However, most of the studies address the use of high doses of corticosteroids [18,19,20,27,28], whether used on a daily basis or in equivalent doses above 5 mg/day of dexamethasone. Doses in these reports range from equivalent doses of dexamethasone of 12 mg/day, 2 days a week to 90 mg/month on a monthly basis, which are actually much higher than the mini-pulse therapy. On the other hand, no studies convenientlyaddress the reduction in SALT scores or the factors associated with the success of the dexamethasone mini-pulse treatment. In our study, it was found that more than 50% patients who undergo dexamethasone treatment can achieve a reduction of 50% in SALT score (SALT 50) at 9 months (Figure 3). Moreover, the improvement in SALT score seems to be independent from the basal SALT of the patients (Table 2). Controlled studies will probably be useful to really know the impact of the treatment on SALT, and the exact response rates that it can achieve.

On the other hand, in line with our results, the early age of onset has been previously associated with a more aggressive course [29]. Regarding the presence of hypothyroidism, similar results have been found in other studies [30], with patients with hypothyroidism having poorer responses in comparison to patients without the disorder. In light of our results, the use of dexamethasone in the treatment of these patients should be avoided or, at least, should be evaluated early to detect a lack of efficacy before the appearance of undesirable adverse effects. Therefore, the prototype patient for mini-pulse therapy would be one with moderate to severe disease, with an extent that is not treatable with topical treatment, preferably with a late disease onset, without hypothyroidism, and without contraindications for the treatment.

Despite that oral minoxidil has been reported as effective in AA [31,32], there is currently a lack of firm evidence to support its use [33]. Moreover, in our study, the co-treatment did not seem to be related to a better improvement in SALT scores, neither after the univariate nor the multivariate analysis, which controlled the possible confusion bias. Therefore, the recommendation to treat these patients with oral minoxidil should be made on an individual basis.

Although response rates (Figure 2) appear to be relatively low when compared to response rates to routine treatments obtained in other skin diseases, such as psoriasis [34], given the great impact on the quality of life of these patients, in most cases, patients are satisfied with the results. However, this statement is based on the subjective impression of the physicians involved in the present study, and should be corroborated by a further detailed quality of life studies in patients treated with dexamethasone. To improve the results of dexamethasone treatment, or when treating the already mentioned patients who are less likely to respond, other therapies, such as JAKi, are probably the alternative of choice [13]. Deeper evaluation of these new drugs is necessary to establish their actual role in the treatment of alopecia areata.

Regarding side effects, the use of reduced doses of corticosteroids, such as the mini-pulses described in this article, is of particular interest. This approach would maintain efficacy and reduce the dose of treatment, thus reducing the appearance of undesirable effects. In addition, it is likely that the use of lower doses will allow the treatment to be used for a longer period of time and to be free of adverse effects.

The main limitation of the present study is the sample size, which could limit the occurrence of significant results. Moreover, as it is a real-life study and not a controlled study, there is a lack of homogeneity in the doses administered. Further studies on the subject would be interesting to better understand the effects of dexamethasone treatment in AA.

## 5. Conclusions

Mini-pulse therapy seems to be an effective and safe therapeutic approach for patients with AA refractory compared to topical therapies, although those patients who had an early age of onset of the disease, or those who are affected by hypothyroidism, seem to have an unsatisfactory response to the treatment. As side effects are related to higher daily and cumulative doses, mini-pulse therapy is of special interest to decrease the rate of side effects and allow for longer treatments.

## Figures and Tables

**Figure 1 jcm-11-01694-f001:**
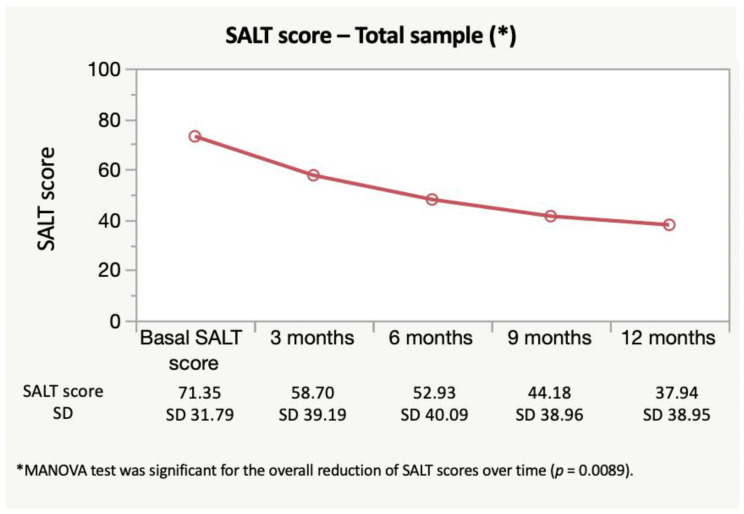
Evolution of SALT scores after the treatment with dexamethasone in the overall sample. A significant decrease of SALT scores was observed.

**Figure 2 jcm-11-01694-f002:**
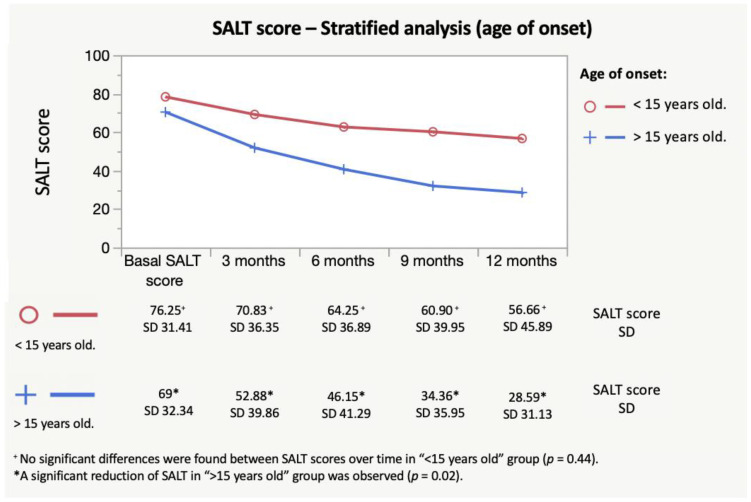
Evolution of SALT scores after the treatment with dexamethasone; stratified analysis by age of onset. Stratified analysis showed that no significant decrease of SALT was found over time in “early onset” group, whereas a significant decrease of SALT scores was present in “late onset” group. No differences were found between both groups in main basal characteristics (Table 2).

**Figure 3 jcm-11-01694-f003:**
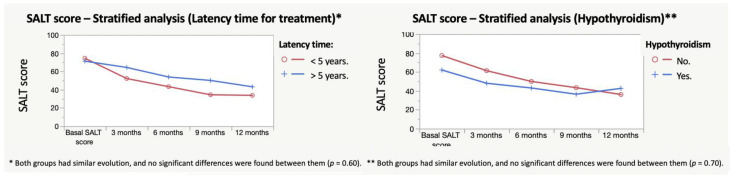
Stratified analyses for latency time for treatment and hypothyroidism did not find significant differences between groups.

**Figure 4 jcm-11-01694-f004:**
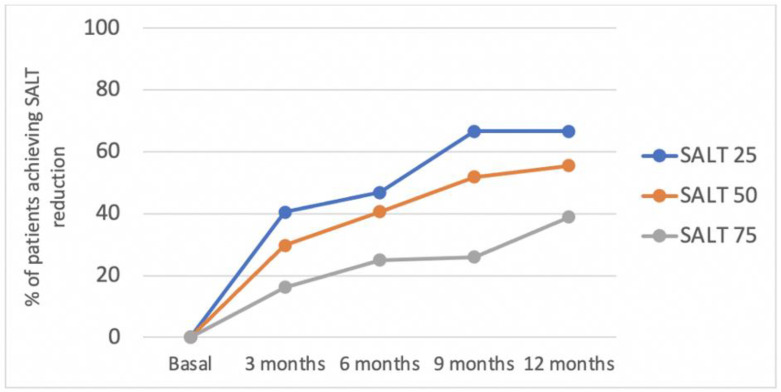
Percentage of patients achieving a reduction of 25%, 50%, and 75% in the SALT score, with respect to baseline (SALT 25, SALT 50, and SALT 75, respectively) 3, 6, 9, and 12 months after dexamethasone treatment.

**Table 1 jcm-11-01694-t001:** Socio-demographic and clinical features of the sample and characteristics of dexamethasone treatment.

Variables All Patients (*n* = 38)					
Socio-demographic Features					
Age (years)	32.28 (SD 16.58)		Age of onset (years)	28.26 (SD 16.74)	
Sex (%)	Male:	42.5% (17/40)	Age of onset (years)	<15	35% (14/40)
	Female:	57.5% (23/40)		>15	65% (26/40)
Age at the start of the treatment (years)	32.28 (SD 16.57)		Evolution time of the disease (years)	5.11 (SD 7.11)	
Previous Treatments					
Topical corticosteroids (%)	100% (40/40)		Topical anthralin (%)	42.5% (17/40)	
Intralesional steroids (%)	60% (24/40)		Oral corticosteroids (%)	52.5% (21/40)	
Topical minoxidil (%)	80% (32/40)		Immunosuppressive agents (%)	17.5% (7/40)	
Comorbidities					
Hypothyroidism (%)	27.5% (11/40)		Type 1 diabetes mellitus (%)	5% (2/40)	
Celiac disease (%)	2.5% (1/40)		Other immunomediated diseases (%)	12.5% (5/40)	
Severity of the Disease					
Basal SALT score	70.77 (SD 31.13)		Eyebrows/eyelashes involvement (%)	56.4% (22/40)	
Treatment Characteristics—All patients received dexamethasone 2 days per week					
Dexamethasone dose (mg/day)	2.71 (SD 1.47)		Treatment time (months)	12.25 (SD 8.89)	
Cumulative dose of dexamethasone (mg)	260.05 (SD 197.92)		Proton-pump inhibitors co-treatment (%)	70% (28/40)	
Calcium/vitamin D co-treatment (%)	82.5% (33/40)		Oral minoxidil co-treatment (%)	27.5% (11/40)	

**Table 2 jcm-11-01694-t002:** Basal characteristics in “early” and “late” onset groups. No differences were found between groups in any of the variables analyzed.

Variables	“Early” Group	“Late” Group	*p*-Value
	(<15 Years Old)	(>15 Years Old)	
Male sex (%)	42.8% (6/14)	42.31% (11/26)	0.97
Dexamethasone dose (mg/day)	2.19 (SD 1.44)	3 (SD 0.28)	0.15
Basal SALT score (%)	73.92 (SD 29.75)	69 (SD 6.41)	0.64
Oral minoxidil co-treatment (%)	14.2% (2/14)	34.62% (9/26)	0.17

**Table 3 jcm-11-01694-t003:** Associated factors analysis for SALT reduction at 3 and 9 months.

Factors	SALT Reduction (3 Months)				SALT Reduction (9 Months)			
	Univariate Analysis		Multivariate Analysis		Univariate Analysis		Multivariate Analysis	
	Difference/Beta	*p*-Value	Beta	*p*-Value	Difference/Beta	*p*-Value	Beta	*p*-Value
Sex	Male: 6.78 (SE 6.74)	0.28	8.33 (SE 4.18) (female sex)	0.05	Male: 25.7 (SE 10.48)	0.63	4.72 (SE 6.58)	0.48
	Female: 16.22 (SE 5.26)				Female: 31.05 (SE 8.03)			
Age	0.38 (SE 0.26)	0.14	-	-	0.69 (SE 0.40)	0.10	-	-
Age of onset	0.41 (SE 0.24)	0.09	0.67 (SE 0.27)	0.02	0.85 (SE 0.36)	0.02	1.06 (SE 0.46)	0.02
Evolution time	−0.03 (SE 0.005)	0.49	-	-	−0.09 (SE 0.06)	0.17	-	-
Family history	Yes: 9.16 (SE 10.4)	0.71	-	-	Yes: 15.40 (SE 14.55)	0.28	-	-
	No: 13.32 (SE 4.6)				No: 32.95 (SE 6.93)			
Basal SALT	0.057 (SE 0.13)	0.67	-	-	0.23 (SE 0.22)	0.32	-	-
Hypothyroidism	Yes: 7.27 (SE 7.66)	0.40	−11.34 (SE 4.87)	0.03	Yes: 25.6 (SE 14.86)	0.76	−12.76 (SE 9.47)	0.19
	No: 14.92 (SE 4.98)				No: 30.73 (SE 7.08)			
Celiac disease	Yes: 0 (SE 25.58)	0.61	-	-	Yes: −2 (SE 32.66)	0.33	-	-
	No: 13 (SE 4.26)				No: 30.92 (SE 6.40)			
Oral minoxidil co-treatment	Yes: 18 (SE 8.05)	0.44	1.85 (SE 4.51)	0.68	Yes: 42.50 (SE 13.20)	0.28	−1.26 (SE 8.46)	0.88
	No: 10.67 (SE 4.89)				No: 7.09 (SE 11.40)			
Dexamethasone dose	0.10 (SE 3.09)	0.97		-	−6.37 (SE 5.49)	0.25	-	-
R^2^	-		0.2500		-		0.2507	

**Table 4 jcm-11-01694-t004:** Side effect-associated factors during dexamethasone treatment.

Factors	Overall Side Effects				Weight Gain			
	Univariate Analysis		Multivariate Analysis		Univariate Analysis		Multivariate Analysis	
	Mean/%	*p*-Value	Beta	*p*-Value	Mean/%	*p*-Value	Beta	*p*-Value
Sex (%)	Male: 46.67%	0.65	0.57 (SE 0.43) (male sex)	0.20	Male: 53.33%	0.08	0.63 (SE 0.42) (male sex)	0.13
	Female: 39.13%				Female: 26.09%			
Age (years)	Yes: 35.09 (SD 4.02)	0.63	-	-	Yes: 30.82 (SE 4.27)	0.41		
	No: 32.55 (SD 3.43)				No: 35.25 (SE 3.26)			
Age of onset (years)	Yes: 31.42 (SD 4.18)	0.33	-	-	Yes: 26.50 (SE 4.52)	0.63		
	No: 25.95 (SD 3.57)				No: 29.29 (SE 3.45)			
Oral minoxidil co-treatment (%)	Yes:36.36%	0.65	-	-	Yes: 36.36%	0.96		
	No: 44.44%				No: 37.04%			
Dexamethasone dose (mg/day)	Yes: 3.38 (SD 0.34)	0.04	1.87 (SE 0.92)	0.04	Yes: 3.11 (SE 0.38)	0.36	0.053 (SE 0.42)	0.89
	No: 2.42 (SD 0.29)				No: 2.66 (SE 0.29)			
Treatment duration (months)	Yes: 14.18 (SD 2.27)	0.27	0.35 (SE 0.38)	0.06	Yes: 15.07 (SE 2.39)	0.15	0.01 (SE 0.11)	0.90
	No: 10.81 (SD 1.93)				No: 10.58 (SE 1.83)			
Cumulative dose of dexamethasone (mg)	Yes: 321 (SD 49.29)	0.17	0.01 (SE 0.008)	0.11	Yes: 363.43 (SE 50.31)	0.02	0.004 (SE 0.005)	0.39
	No: 229.8 (SD 42.03)				No: 212.67 (SE 38.42)			
R^2^	-		0.2048				0.1666	

## Data Availability

The data presented in this study are available on reasonable request from the corresponding author.

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
