# Peer review of "Alopecia Areata and Dexamethasone Mini-Pulse Therapy, A Prospective Cohort: Real World Evidence and Factors Related to Successful Response"

_jcm, 2022, doi:10.3390/jcm11061694_

Round 1
Reviewer 1 Report
This paper provides a possible beneficial efficacy against alopecia areata. I agree almost all of data in this paper. I think it is better to highlight which population or characteristics will become a candidate of mini-pulse therapy among various therapeutic options.
Author Response
Dear Reviewer,
Thank you very much for your comments, as they allow us to improve the scientific quality of our work.
Some text regarding this issue has been added to the discussion, to clarify the prototype patient for mini-pulse therapy treatment
Reviewer 2 Report
Just a few comments:
The most common spelling I found was diphencyprone.
I was a little intrigued by the fact that the duration of the AA did not influence the results. Am I wrong?
Was there any relationship between age or duration of treatment with dexamethasone in relation to osteoporosis?
In short, the article is good, well written, brings relevant contributions to dermatological practice.
Author Response
Dear Reviewer,
Thank you very much for your comments, as they allow us to improve the scientific quality of our work.
1) The spelling mistake has been corrected.
2) In our study, disease duration was found to be not related to the response to the treatment. However, it should be noted that the relatively small sample size may have limited the occurrence of statistically significant differences.
3) This point is very interesting. We have added an univariate analysis in the results section addressing this issue: "Regarding osteopenia osteoporosis, patients who suffered from osteopenia or osteoporosis during dexamethasone treatment (12.5%, 5/40) were preferably female (p=0.01), had longer treatments (p=0.001) and had higher cumulative doses of dexamethasone (p=0.03) when compared to patients who did not suffered from it."
Reviewer 3 Report
You stated that "Oral corticosteroids, 49 administered in pulses have shown to be effective in the management of AA (15,16), alt-50 hough most studies address the issue of high dose systemic corticosteroids (17–20), which 51 can lead to several adverse effects (21). " please add side effects here
- "Dexamethasone doses <5mg/day, two days per week." do you have standardized dose in patients <15 years?
-
3.2. Characteristics of the treatment with dexamethasone: 143
All patients were treated with mini-pulses of dexamethasone (table 1). Mean dexa-144 methasone dose was 2.72 mg/day PER WEEK?(SD 1.48) with a mean total treatment time of 12.22 145 months (SD 8.89).
Author Response
Dear Reviewer,
Thank you very much for your comments, as they allow us to improve the scientific quality of our work.
1) Side effects of corticoid treatment has been added in the suggested place.
2) As this is a real-world evidence study, drug doses were prescribed according to standard clinical practice, so there were no standardized doses for this particular group. However, it is true that the doses in this group were usually much lower than those of the adults, usually less than 2 mg/day.
3) This sentence has been corrected to make it easier to understand.
Round 2
Reviewer 1 Report
This paper is improved well. I do not have any additional suggestion. Good luck!